# Mental health and health-related quality of life among adults with osteoarthritis: A national population-based study

**Monira Alwhaibi**[iD]*◉, **Tariq M. Alhawassi**◉

Department of Clinical Pharmacy, College of Pharmacy, King Saud University, Riyadh, Saudi Arabia

◉ These authors contributed equally to this work.
* malwhaibi@ksu.edu.sa

## Abstract

### Background

Adults with osteoarthritis are vulnerable to mental health disorders, which may have a significant impact on their Health-Related Quality of Life (HRQoL). Thus, the purpose of this study was to look at the relationship between anxiety and depression and HRQoL in adults with osteoarthritis.

### Methods

Adult patients diagnosed with osteoarthritis who were 18 years of age or older were identified using data from the Medical Expenditure Panel Survey for the years 2018–2021. HRQoL was computed using the veteran's RAND 12-item health survey which has two domains the Physical and Mental Component Summary (PCS & MCS) ratings. The link between anxiety and depression and HRQoL in individuals with osteoarthritis was examined using multivariable linear regression after the adjustment for a variety of covariates.

### Results

Of the 3,658 individuals with osteoarthritis that were identified, 12.0% experienced depression, 12.5% had anxiety, and 9.9% had both illnesses. The PCS and MCS mean scores of the HRQoL were lowest for persons with comorbid depression and anxiety compared to adults with osteoarthritis only. Moreover, from the adjusted regression analysis, adults with osteoarthritis who had depression (MCS: adjusted $\beta = -5.408$, $p < 0.001$), anxiety (MCS: adjusted $\beta = -3.485$, $p < 0.001$), or both depression and anxiety (MCS: adjusted $\beta = -10.348$, p-value<0.0001) had significantly lower HRQoL MCS scores compared to those with osteoarthritis only. Those who were physically active and employed had notably better scores on the PCS and MCS than those who were not.

provided the original author and source are credited.

**Data availability statement:** The dataset used in this study is available from the MEPS database at this URL: https://meps.ahrq.gov/mepsweb/data_stats/download_data_files.jsp.

**Funding:** This research was supported by the Ongoing Research Funding Program, (ORF-2025-1128), King Saud University, Riyadh, Saudi Arabia.

**Competing interests:** The author affirms that there are no conflicts of interest related to the publication of this paper.

**Abbreviations:** HRQoL, health-related quality of life; MCS, mental health component score; PCS, physical health component score.

## Conclusions

This nationally representative sample has shed light on the relationships between anxiety, depression, and a low HRQoL in adults with osteoarthritis. The study also revealed the impact of socioeconomic factors such as education level and income on HRQoL, and the benefits of employment and regular exercise. Importantly, the findings underscore the need for lifestyle modifications to reduce anxiety and depression in individuals with osteoarthritis. They also highlight the practical implications for healthcare planning and resource allocation, providing valuable insights for policymakers and healthcare professionals.

## 1. Introduction

Osteoarthritis (OA) is one of the most prevalent disabling illnesses globally and has been shown to have serious adverse effects on health [1]. It was estimated that OA affected around six hundred million individuals worldwide in 2020, representing 7.6% of the world's population [2]. Moreover, OA cases are expected to rise by nearly 50% for the hand, 75% for the knee and the hip, and up to 95% increase in OA prevalence for other types by 2050 as compared to 2020 [2]. Nowadays, OA is the 14th leading cause of age-standardized years lived with disability (YLDs) globally and has a substantial burden on individuals characterized by pain, limitations of activity, and diminished quality of life [2]. In addition to the direct health-related burden, OA has a significant financial burden and detrimental effects on a person's psychological health and well-being, work participation, sleep, mobility, and diminished health-related quality of life (HRQoL) [1,3–10]. Therefore, individuals with OA have three times more potential than individuals in the general population to experience anxiety and depression, which could be attributed to pain severity, poor function, number of OA sites, immunological inflammation, and structural alterations in the brain [11–13].

Anxiety and depression can increase the risk of adverse clinical outcomes, such as OA illness burden and morbidity, and also result in increased healthcare costs and health service utilization, with higher healthcare provider visits, emergency room visits, and hospitalizations [12,14–16]. Besides, depression was associated with lower HRQoL for individuals with OA [17,18]. HRQoL is a crucial outcome measure for evaluating the overall effects of long-term illnesses like OA, including a person's social, emotional, and physical well-being [19].

With the aforementioned risk of OA on patients' psychological well-being, maintaining a high quality of life in terms of health despite the difficulties posed by OA is a primary objective for adult OA sufferers and their healthcare providers. Therefore, understanding the causes underlying a lower HRQoL becomes essential, especially when considering mental health. Although prior studies have explored the impact of mental health on health-related quality of life (HRQoL) in individuals with osteoarthritis (OA), many face methodological and contextual limitations that restrict their applicability to broader populations or policymaking. [6,12,15,17–21]. For example, Chang et al. (2023), Lee et al. (2020), and Kim et al. (2019) used data from the Korean

National Health and Nutrition Examination Survey (KNHANES), but the diagnoses of OA and mental health conditions were self-reported without clinical validation, raising concerns about measurement accuracy. Tsuji et al. (2019) employed Internet-based survey methods via the National Health and Wellness Survey (NHWS), introducing potential selection bias, especially underrepresentation of older adults. Barbosa et al. (2022) conducted their analysis on a small sample (n = 107), limiting statistical power and generalizability. Tavares et al. (2022), reported heavy reliance on subjective measures (e.g., visual analog scales) and gender imbalance further limit the robustness of conclusions.

These cumulative limitations from small or biased samples to omitted confounders underscore the need for comprehensive, population-based analyses using validated instruments and broader covariate adjustment. Our study addresses these gaps by leveraging nationally representative data from the Medical Expenditure Panel Survey (MEPS, 2018–2021), which includes clinically coded OA diagnoses (ICD-10-CM), validated HRQoL measures (VR-12), and extensive information on social determinants, comorbidities, and health behaviors. This approach allows us to offer more generalizable and clinically relevant insights into how depression and anxiety affect HRQoL in adults with OA. Therefore, this led us to conduct this research, which aimed at evaluating the association between depression, anxiety, and the HRQoL among adults with OA using a sample from a representative United States (US) database and adding more insightful knowledge to the currently available research in this significant field.

## 2. Methods

### 2.1. Study design and data

This study employed a retrospective longitudinal design using data from the 2020–2021 cycle of the Medical Expenditure Panel Survey (MEPS), a nationally representative survey of the U.S. noninstitutionalized civilian population conducted by the Agency for Healthcare Research and Quality (AHRQ). MEPS utilizes an overlapping panel structure in which data are collected from households over two calendar years using a stratified, multistage probability sampling design. Data collection is conducted through multiple rounds of computer-assisted personal interviews. No ethical review or approval was needed as we have utilized a publicly accessible, the Medical Expenditure Panel Survey (MEPS) database. Medical Expenditure Panel Survey (MEPS) database is publicly accessible secondary data, therefore, patient consent was waived for this research.MEPS provides extensive information on a range of variables, including demographic characteristics, chronic health conditions, healthcare utilization and expenditures, access to and satisfaction with care, health insurance coverage, income, and employment status. Medical conditions reported by respondents are based on health issues experienced during the data collection period. These self-reported conditions are recorded verbatim during the interviews and subsequently coded by professional coders into standardized International Classification of Diseases, Tenth Revision, Clinical Modification (ICD-10-CM) codes.

### 2.2. Study population

According to the study's inclusion criteria, an adult must be between 18 and 64 years old, have a diagnosis of OA, alive within the study period, and have no missing data on HRQoL. The International Classification of Diseases, tenth Revision, Clinical Modification (ICD-10-CM) clinical diagnosis codes were used to identify OA from MEPS data (M16: osteoarthritis of the hip, M17: osteoarthritis of the knee, M19: other and unspecified osteoarthritis) [22].

### 2.3. Measures

**2.3.1. Outcome: Health-related quality of life (HRQoL).** To evaluate the HRQoL, adults who participated in the MEPS filled out a questionnaire using a reliable and valid instrument, the Veterans RAND 12 Item Health Survey (VR-12©) [23–25]. VR-12 is a 12-item patient-reported outcome measure that evaluates eight domains of health: general health, physical functioning, role physical, bodily pain, vitality, role emotional, mental health, and social functioning [23].

The VR-12 has two parts: 1/ Physical Component Summary (PCS), which included responses from the general health, physical functioning, role physical, and bodily pain domains, and 2/ Mental Component Summary (MCS), which included responses from role emotional, vitality, mental health, and social functioning domains [26]. Scores range from 0 to 100, with higher PCS and MCS scores as better physical and mental HRQoL indicators.

**2.4.2. Independent variables.** The primary independent variable of interest in this study was the osteoarthritis (OA) group classification, defined as mutually exclusive categories: OA only, OA with anxiety, OA with depression, and OA with both anxiety and depression. Additional covariates were selected based on well-established evidence of their association with health-related quality of life (HRQoL) in adults with OA. These include sociodemographic variables (age, gender, education, income, employment, marital status, and region). These are key social determinants of health known to influence both physical and mental HRQoL [27–33]. A health behavior (physical activity) has been selected as regular exercise is strongly associated with improved physical function and mental well-being in OA populations. Perceived health status has been included as self-rated health is a robust predictor of HRQoL. Chronic comorbidities (e.g., hypertension, diabetes, asthma, COPD, GERD), are frequently observed in individuals with OA and contribute significantly to the overall disease burden and quality of life. Health insurance coverage, included as a proxy for access to healthcare services, which can directly affect HRQoL outcomes.

## 2.5. Statistical analyses

Descriptive statistics were employed to show the study sample's baseline characteristics, including mean, standard deviation, frequencies, and percentages. The differences in individual characteristics between the OA groups were evaluated using chi-square tests. The mean variances in HRQoL by OA groups were determined using ANOVA followed by Tukey's post hoc test for multiple comparisons to identify which specific groups differed. To assess the link between OA groups and HRQoL, we used multivariable linear regression and adjusted for all independent covariates. Variables were included in the model based on their theoretical relevance, clinical importance, and prior literature supporting their association with health-related quality of life (HRQoL) in OA patients [27–33]. Additionally, a bivariate analysis was performed and only variables with a likelihood ratio p-value of <0.05 were considered for the multivariable regression model. All the underlying assumptions of the linear regression model have been evaluated to ensure that they are satisfied by the regression model and are met by the variables. Those assumptions include correct model specification, normality, homoscedasticity, no autocorrelation, and no multicollinearity. After checking the underlying assumptions, we performed the linear regression model. All estimations in the statistical analysis considered the complicated survey design of the MEPS by integrating variance adjustment weights (strata and primary sample unit) from the MEPS with person-level weights. The Statistical Analysis System, SAS 9.4 (SAS Institute Inc., Cary, NC, USA), was used to analyze the data.

## 3. Results

### 3.1. Characteristics of the study sample

Table 1 lists the characteristics of the study sample, 3,658 adults with OA; the majority of the study sample was women (62.3%), aged 50–64 (72.0%), White (73.3%), and had higher than high school (87.8%). About 12.0% of adults with OA also experienced depression, 12.5% had anxiety, and 9.9% had both illnesses.

Women with OA showed significantly higher rates of depression (13.6% vs. 9.3%), anxiety (14.7% vs. 9.0%), and comorbid anxiety and depression (12.0% vs. 6.5%, p-value = <0.0001) when compared to men. Furthermore, unemployed OA patients had significantly higher rates of depression (16.2% vs. 9.3%) and comorbid anxiety and depression (13.7% vs. 7.5%, p-value <0.0001) than employed OA patients. OAs adults with diabetes, asthma, COPD, and GERD had significantly higher rates of comorbid depression and anxiety than those without these comorbidities (p-value <0.0001).

**Table 1. Socio-economic and health-related characteristic of study participants across osteoarthritis groups.**

| | Total Sample | | OA Only | | OA & Depression | | OA & Anxiety | | OA & Depression & Anxiety | | P-value* |
|---|---|---|---|---|---|---|---|---|---|---|---|
| | N | Wt.% | N | Wt.% | N | Wt.% | N | Wt.% | N | Wt.% | |
| **All** | 3,658 | 100.0 | 2361 | 65.6 | 473 | 12.0 | 429 | 12.5 | 395 | 9.9 | |
| **Age in years** | | | | | | | | | | | |
| 22-39 | 327 | 9.7 | 187 | 54.4 | 31 | 10.8 | 52 | 16.8 | 57 | 18.0 | <0.0001 |
| 40-49 | 648 | 18.2 | 384 | 62.4 | 87 | 12.7 | 90 | 13.4 | 87 | 11.5 | |
| 50-64 | 2683 | 72.0 | 1790 | 67.9 | 355 | 12.0 | 287 | 11.7 | 251 | 8.4 | |
| **Gender** | | | | | | | | | | | |
| Women | 2405 | 62.3 | 1437 | 59.8 | 336 | 13.6 | 319 | 14.7 | 313 | 12.0 | <0.0001 |
| Men | 1253 | 37.7 | 924 | 75.2 | 137 | 9.3 | 110 | 9.0 | 82 | 6.5 | |
| **Race/ethnicity** | | | | | | | | | | | |
| White | 2304 | 73.3 | 1412 | 63.5 | 318 | 12.4 | 305 | 13.8 | 269 | 10.2 | 0.011 |
| African American | 658 | 12.5 | 482 | 73.6 | 66 | 9.5 | 62 | 8.5 | 48 | 8.4 | |
| Latino | 488 | 8.8 | 336 | 72.1 | 58 | 11.3 | 42 | 8.1 | 52 | 8.5 | |
| Others | 208 | 5.4 | 131 | 64.9 | 31 | 13.0 | 20 | 11.1 | 26 | 11.0 | |
| **Marital status** | | | | | | | | | | | |
| Married | 1749 | 54.8 | 1232 | 71.0 | 190 | 10.3 | 199 | 12.2 | 128 | 6.5 | <0.0001 |
| Widow/Sep/Div | 1250 | 30.2 | 712 | 57.6 | 200 | 14.1 | 166 | 13.9 | 172 | 14.4 | |
| Never married | 659 | 15.0 | 417 | 62.1 | 83 | 13.7 | 64 | 11.0 | 95 | 13.2 | |
| **Education level** | | | | | | | | | | | |
| <High School | 197 | 3.4 | 133 | 70.8 | 24 | 12.2 | 22 | 9.8 | 18 | 7.2 | 0.015 |
| High School | 395 | 8.0 | 240 | 61.7 | 54 | 13.2 | 38 | 8.7 | 63 | 16.5 | |
| > High School | 3033 | 87.8 | 1965 | 65.7 | 392 | 11.9 | 365 | 13.0 | 311 | 9.4 | |
| **Region** | | | | | | | | | | | |
| Northeast | 589 | 16.9 | 385 | 68.3 | 70 | 9.4 | 57 | 10.1 | 77 | 12.2 | 0.356 |
| Mid-west | 939 | 26.3 | 572 | 63.4 | 145 | 13.9 | 113 | 12.2 | 109 | 10.4 | |
| South | 1444 | 38.8 | 946 | 64.9 | 169 | 12.5 | 175 | 13.1 | 154 | 9.5 | |
| West | 686 | 18.0 | 458 | 67.7 | 89 | 10.5 | 84 | 14.1 | 55 | 7.7 | |
| **Employment** | | | | | | | | | | | |
| Employed | 1942 | 60.8 | 1399 | 72.0 | 192 | 9.3 | 198 | 11.3 | 153 | 7.5 | <0.0001 |
| Not employed | 1716 | 39.2 | 962 | 55.7 | 281 | 16.2 | 231 | 14.4 | 242 | 13.7 | |
| **Poverty status** | | | | | | | | | | | |
| Poor | 814 | 15.8 | 441 | 50.7 | 127 | 15.8 | 114 | 15.5 | 132 | 18.0 | <0.0001 |
| Near Poor | 723 | 16.0 | 423 | 57.7 | 112 | 16.8 | 93 | 13.3 | 95 | 12.2 | |
| Middle Income | 911 | 24.8 | 606 | 66.4 | 123 | 12.7 | 101 | 12.4 | 81 | 8.6 | |
| High Income | 1210 | 43.5 | 891 | 73.5 | 111 | 8.4 | 121 | 11.2 | 87 | 6.8 | |
| **Health Insurance** | | | | | | | | | | | |
| Private | 2098 | 67.0 | 1507 | 71.9 | 227 | 9.8 | 210 | 11.2 | 154 | 7.1 | <0.0001 |
| Public | 1415 | 29.4 | 755 | 51.3 | 227 | 16.6 | 207 | 15.7 | 226 | 16.4 | |
| Uninsured | 145 | 3.6 | 99 | 65.5 | 19 | 14.0 | 12 | 11.2 | 15 | 9.3 | |
| **Rx Insurance** | | | | | | | | | | | |
| Rx insurance | 1850 | 59.4 | 1347 | 72.8 | 185 | 9.3 | 187 | 11.1 | 131 | 6.8 | <0.0001 |
| No Rx insurance | 1808 | 40.6 | 1014 | 55.1 | 288 | 15.9 | 242 | 14.5 | 264 | 14.4 | |
| **General health** | | | | | | | | | | | |
| Excellent/very good | 1002 | 31.3 | 770 | 77.2 | 100 | 8.5 | 96 | 10.9 | 36 | 3.4 | <0.0001 |
| Good | 1302 | 36.8 | 876 | 66.8 | 154 | 11.7 | 146 | 11.9 | 126 | 9.6 | |
| Fair/poor | 1354 | 31.9 | 715 | 52.8 | 219 | 15.8 | 187 | 14.8 | 233 | 16.6 | |

*(Continued)*

**Table 1.** (Continued)

| | Total Sample | | OA Only | | OA & Depression | | OA & Anxiety | | OA & Depression & Anxiety | | P-value[*] |
|---|---|---|---|---|---|---|---|---|---|---|---|
| | N | Wt.% | N | Wt.% | N | Wt.% | N | Wt.% | N | Wt.% | |
| **Physical activity** | | | | | | | | | | | |
| 3/week | 1496 | 43.1 | 1054 | 71.2 | 169 | 9.8 | 156 | 12.0 | 117 | 7.0 | <0.0001 |
| No exercise | 2158 | 56.8 | 1305 | 61.4 | 302 | 13.6 | 273 | 12.9 | 278 | 12.1 | |
| **Heart Disease** | | | | | | | | | | | |
| Yes | 422 | 10.5 | 235 | 58.6 | 63 | 14.7 | 62 | 13.8 | 62 | 12.8 | 0.111 |
| No | 3236 | 89.5 | 2126 | 66.4 | 410 | 11.7 | 367 | 12.4 | 333 | 9.5 | |
| **Hypertension** | | | | | | | | | | | |
| Yes | 1820 | 45.9 | 1107 | 62.6 | 272 | 14.2 | 236 | 12.9 | 205 | 10.2 | 0.017 |
| No | 1838 | 54.1 | 1254 | 68.2 | 201 | 10.1 | 193 | 12.2 | 190 | 9.6 | |
| **Diabetes** | | | | | | | | | | | |
| Yes | 848 | 21.3 | 471 | 56.0 | 149 | 17.5 | 99 | 12.7 | 129 | 13.7 | <0.0001 |
| No | 2810 | 78.7 | 1890 | 68.2 | 324 | 10.5 | 330 | 12.5 | 266 | 8.9 | |
| **Hyperlipidemia** | | | | | | | | | | | |
| Yes | 1359 | 35.6 | 788 | 60.0 | 208 | 14.3 | 185 | 14.1 | 178 | 11.5 | 0.004 |
| No | 2299 | 64.4 | 1573 | 68.7 | 265 | 10.7 | 244 | 11.6 | 217 | 9.0 | |
| **Asthma** | | | | | | | | | | | |
| Yes | 706 | 17.5 | 363 | 52.5 | 105 | 14.7 | 103 | 15.3 | 135 | 17.6 | <0.0001 |
| No | 2952 | 82.5 | 1998 | 68.4 | 368 | 11.4 | 326 | 11.9 | 260 | 8.3 | |
| **COPD** | | | | | | | | | | | |
| Yes | 432 | 10.9 | 214 | 49.7 | 61 | 15.1 | 67 | 16.1 | 90 | 19.1 | <0.0001 |
| No | 3226 | 89.1 | 2147 | 67.5 | 412 | 11.6 | 362 | 12.1 | 305 | 8.8 | |
| **GERD** | | | | | | | | | | | |
| Yes | 719 | 18.4 | 382 | 53.2 | 111 | 15.6 | 105 | 15.0 | 121 | 16.2 | <0.0001 |
| No | 2939 | 81.6 | 1979 | 68.4 | 362 | 11.2 | 324 | 12.0 | 274 | 8.5 | |

[*]Significant differences in osteoarthritis (OA) groups were represented by chi-square tests of likelihood ratio p-value.

COPD: Chronic obstructive pulmonary disease; GERD: Gastroesophageal reflux disease; Wt: weighted; Rx: Medication; Wid./Div./Sep.: widowed, divorced, and separated

## 3.2. Health-related quality of life and osteoarthritis groups

A significant difference in the PCS and MCS scores was found between OA groups using ANOVA (Table 2). For example, OA adults with both depression and anxiety had a significantly lower mean PCS score (Mean = 36.84, SE = 0.78) and MCS score (Mean = 37.47, SE = 0.74) compared to other groups (Mean MSC score was 51.20 for OA only, 42.73 for OA and depression, and 45.71 for OA and anxiety). ANOVA test, which had shown a significant difference in all groups as P < 0.0001, was followed by Tukey's Post Hoc test for multiple comparisons, which found that regarding the PCS score, adults with OA only have a significantly higher mean PCS score as compared to the other groups (OA and anxiety, OA and depression, and OA and anxiety and depression). Also, adults with OA and anxiety/depression have a significantly lower mean PCS score as compared to all groups. Regarding the MCS score, adults with OA only have a significantly higher mean MCS score as compared to the other groups. Also, adults with OA and anxiety/depression have a significantly lower mean MCS score as compared to all other groups. In summary, patients with only OA without any mental health disorder have the highest level of PCS and MCS scores (quality of life).

**Table 2. Health-related Quality of Life Weighted Means Scores by Osteoarthritis (OA) Groups.**

| | Total Sample | | OA Only | | OA & Depression | | OA & Anxiety | | OA & Depression & Anxiety | | |
|---|---|---|---|---|---|---|---|---|---|---|---|
| | Mean | SD | Mean | SE | Mean | SE | Mean | SE | Mean | SE | p-value |
| **HRQoL** | | | | | | | | | | | |
| **PCS** | 39.91 | 12.43 | 43.12 | 0.32 | 37.39 | 0.79 | 40.91 | 0.78 | 36.84 | 0.78 | <0.0001 |
| **MCS** | 47.21 | 11.00 | 51.20 | 0.22 | 42.73 | 0.61 | 45.71 | 0.61 | 37.47 | 0.74 | <0.0001 |

Mean differences between osteoarthritis groups were analyzed using an ANOVA test.

HRQoL: Health-related Quality of Life; SE: Standard Error; SD: Standard Deviation; Sig: Significance.

### 3.3. Health-related quality of life among osteoarthritis groups from linear regression analysis

Table 3 shows the adjusted relationship between OA groups and HRQoL. After controlling for all independent variables, OA adults with depression (MCS: β = −6.124), anxiety (MCS: β = −3.424), and both depression and anxiety (MCS: β = −10.317) had a significantly lower HRQoL mental health domain compared to those with OA only (p-value<0.0001).

OA patients, who were employed, lived in Northeast and Mid-west regions, perceived their physical health to be excellent, and exercised regularly tended to have a better quality of life. For example, OA adults who were employed have a significantly higher HRQoL in both physical health domain (PCS: β = 4.588, p-value<0.0001) and mental health domain (MCS: β = 2.858, p-value<0.0001) compared to those who were not employed.

Several factors were found to negatively impact health-related quality of life (HRQoL), including younger age, education levels, lower income, and hypertension. For instance, adults with lower income, high school education, and hypertension had significantly lower physical health domain compared to those with high income, less than high school education, and those with hypertension (hypertension PCS: β = −1.840, p-value <0.0001). Additionally, younger adults (aged 22–39) reported a significantly lower mental health scores (MCS: β = −3.119, p-value <0.0001) compared to those aged 50–64.

## 4. Discussion

The findings of this study provide robust evidence that comorbid depression and anxiety are significantly associated with reduced health-related quality of life (HRQoL) among adults with osteoarthritis (OA). By leveraging a nationally representative sample of U.S. adults, our study addressed important gaps in the existing literature, offering greater insight into how these mental health conditions affect both physical and mental HRQoL domains. Interpreting our findings in the context of international research further strengthens their relevance. Consistent patterns have been observed in studies conducted in Korea, Japan, and Brazil, where comorbid anxiety and depression were also shown to adversely impact HRQoL in individuals with OA. For example, data from the Korea National Health and Nutrition Examination Survey identified psychiatric comorbidities as strong predictors of diminished HRQoL among OA populations. Similarly, Tsuji et al. in Japan reported that depressive symptoms were significantly associated with poorer self-rated health and functional outcomes in OA patients. The alignment of our findings with these studies reinforces the global burden of mental comorbidities in OA and supports the external validity of our results.

In our study, we compared four groups of adults with OA in terms of their quality of life: 1) those with OA only, 2) those with OA and depression, 3) those with OA and anxiety, and 4) those with OA and both depression and anxiety. By comparing these groups, we observed that those with anxiety and depression had worse HRQoL than adults with any of these comorbidities only, or without any of them. Adults with OA and both depression and anxiety have a lower PCS score and MCS score compared to those with OA only (PCS score of 36.84 vs. 43.12, MCS score of 37.47 vs. 51.20), presence of anxiety and depression led to a nearly 6-point reduction in PCS score and a 14-point reduction in MCS score among adults with OA. In our adjusted analysis, the β-coefficient for those with both depression and anxiety (MCS: β = −10.317)

**Table 3. Adjusted multivariable linear regressions on HRQoL among adults with osteoarthritis.**

| | Health-Related Quality of Life | | | | | |
| | Physical Health Summary (PCS) | | | Mental Health Summary (MCS) | | |
| | Regression Coefficients | SE | Sig. | Regression Coefficients | SE | Sig. |
|---|---|---|---|---|---|---|
| **Osteoarthritis (OA) Group** | | | | | | |
| OA & depression | −1.042 | 0.716 | | −6.124 | 0.810 | *** |
| OA & anxiety | 1.316 | 0.736 | | −3.424 | 0.617 | *** |
| OA & depression & anxiety | −0.759 | 0.850 | | −10.317 | 0.855 | *** |
| OA only (Ref.) | | | | | | |
| **Age in years** | | | | | | |
| 22-39 | 2.398 | 1.029 | * | −3.119 | 0.847 | *** |
| 40-49 | 0.575 | 0.776 | | −2.002 | 0.682 | ** |
| 50-64 (Ref.) | | | | | | |
| **Gender** | | | | | | |
| Men | 0.198 | 0.531 | | −0.182 | 0.449 | |
| women (Ref.) | | | | | | |
| **Race/ethnicity** | | | | | | |
| African American | 1.133 | 0.728 | | 0.832 | 0.631 | |
| Latino | 1.245 | 0.839 | | −1.319 | 0.807 | |
| Others | −0.496 | 1.091 | | −1.404 | 1.108 | |
| White (Ref.) | | | | | | |
| **Marital status** | | | | | | |
| Married | −0.519 | 0.620 | | 0.000 | 0.561 | |
| Widow/Separated/Divorce | 0.471 | 0.693 | | 0.591 | 0.642 | |
| Never married (Ref.) | | | | | | |
| **Education level** | | | | | | |
| > High School | −2.203 | 1.385 | | 1.469 | 1.433 | |
| High School | −2.906 | 1.461 | * | 2.815 | 1.527 | |
| <High School (Ref.) | | | | | | |
| **Region** | | | | | | |
| Northeast | 1.448 | 0.608 | * | 0.029 | 0.665 | |
| Mid-west | 3.023 | 0.749 | *** | −0.321 | 0.933 | |
| South | 1.058 | 0.604 | | −0.053 | 0.682 | |
| West (Ref.) | | | | | | |
| **Employment** | | | | | | |
| Employed | 4.588 | 0.673 | *** | 2.858 | 0.491 | *** |
| Not employed (Ref.) | | | | | | |
| **Poverty status** | | | | | | |
| Poor | −2.115 | 0.857 | * | −1.205 | 0.853 | |
| Near Poor | −0.684 | 0.842 | | −1.058 | 0.800 | |
| Middle Income | −0.981 | 0.647 | | 0.400 | 0.577 | |
| High Income (Ref.) | | | | | | |
| **Health Insurance** | | | | | | |
| Private | 3.034 | 1.590 | | 0.822 | 1.253 | |
| Public | −1.696 | 1.702 | | 0.711 | 1.233 | |
| Uninsured (Ref.) | | | | | | |

*(Continued)*

 

**Table 3.** (Continued)

| | Health-Related Quality of Life | | | | | |
| | Physical Health Summary (PCS) | | | Mental Health Summary (MCS) | | |
| | Regression Coefficients | SE | Sig. | Regression Coefficients | SE | Sig. |
|---|---|---|---|---|---|---|
| **Rx Insurance** | | | | | | |
| Rx insurance | 0.863 | 0.833 | | −0.957 | 0.766 | |
| No Rx insurance (Ref.) | | | | | | |
| **General health** | | | | | | |
| Excellent/very good | 11.075 | 0.710 | *** | 6.215 | 0.626 | *** |
| Good | 6.175 | 0.613 | *** | 4.004 | 0.616 | *** |
| Fair/poor (Ref.) | | | | | | |
| **Physical activity** | | | | | | |
| 3/week | 3.382 | 0.469 | *** | 0.918 | 0.439 | * |
| No exercise (Ref.) | | | | | | |
| **Heart Disease** | | | | | | |
| Yes | −0.486 | 0.776 | | −0.729 | 0.784 | |
| **Hypertension** | | | | | | |
| Yes | −1.840 | 0.494 | *** | −0.089 | 0.519 | |
| **Diabetes** | | | | | | |
| Yes | −0.928 | 0.549 | | −0.079 | 0.572 | |
| **Asthma** | | | | | | |
| Yes | −1.338 | 0.705 | | −1.175 | 0.703 | |
| **COPD** | | | | | | |
| Yes | −0.853 | 0.791 | | −0.443 | 1.066 | |
| **GERD** | | | | | | |
| Yes | −0.517 | 0.645 | | −1.162 | 0.641 | |

Asterisks denote statistical significance from multivariable linear regression on health-related quality of life among adults with osteoarthritis,

***$P < 0.001$;

**$0.001 \leq P < 0.01$;

*$0.01 \leq P < .05$

>: Greater than; <: Less than

COPD: Chronic obstructive pulmonary disease; GERD: Gastroesophageal reflux disease; Rx: Medication; Ref: reference group; SE: Standard Error; Sig: Significance.

indicates that having anxiety and depression comorbidity, as opposed to no mental comorbidity in adults with OA is associated with a 10-unit reduction in HRQoL score in the MCS domain. These findings indicate the incremental effect of having anxiety and depression on health status and reflect a greater disease burden of these mental illnesses. Our analysis's findings concur with those of other previously published research. For example, a study using data from the Korea National Health and Nutrition Examination Survey revealed lower HRQoL in OA adults with psychiatric disease [17]. As nearly one in ten adults with OA in our study have depression or anxiety, these findings suggest the need for OA point-of-care anxiety and depression screening [34], since they are consistently underdiagnosed and neglected. Many general practitioners and primary care providers often overlook these comorbidities because they either only address the physical aspects of OA or neglect to evaluate their patients' psychological well-being entirely [35]. Understanding these comorbidities is crucial because they can alter OA management and progression. The National Institute for Health and

Care Excellence (NICE) has provided guidelines for practitioners to guide them in the holistic evaluation of patients with OA [36]. Therefore, policies and initiatives must promote and support coordinated care that covers OA and mental health management to improve their quality of life, reduce the disease burden, and fill the mental health care gap [37].

Additionally, the results of this study demonstrated that other risk factors including income, education levels, and coexisting hypertension, have an unneglectable impact on HRQoL among adults with OA. In fact, in our descriptive analysis we found that those with high school education level, low income, and those with chronic illnesses have a higher prevalence of depression and anxiety. Previous studies that involved adults in general and adults with OA reported that these are important determinants of HRQoL [6,18,27–33]. The negative association between having high school education and physical health (PCS domain of HRQoL), compared to less than a high school education, may be explained by the higher prevalence of depression and anxiety comorbidity from bivariate analysis in this group, as mental health conditions can exacerbate physical health issues through poor health behaviors. A recently published data from the Epidemiology of Chronic Diseases cohort study, a 10-year follow-up for 983 hip and knee OA patients reported that multimorbidity was associated with a higher risk of poor physical function and HRQoL trajectories [38]. Besides, in earlier studies, chronic illnesses are linked to mental health and are associated with a reduced HRQoL in their sufferers because of their chronic nature and management [39–41]. Given that enhancing HRQoL is seen as a noteworthy outcome and a predictor of disease management, it emphasizes the need for comprehensive care that addresses OA management and physical and mental comorbidities. Recognizing income, education, and chronic illnesses as essential factors in adults with OA can guide healthcare providers in identifying of OA individuals who may be at higher risk of experiencing poor HRQoL and tailoring interventions accordingly. For example, targeting those with low education and income with additional preventative strategies can help in mitigate the negative impact of such risk factors and prevent or minimize impairment in the quality of life of individuals with OA.

Furthermore, this study found that OA adults who were employed, lived in Northeast and Mid-west regions, perceived their physical health to be excellent, exercise regularly had higher HRQoL. Employment was particularly associated with improved physical and mental well-being, likely due to financial stability, social engagement, and a sense of purpose. Additionally, individuals with higher incomes generally experienced better health outcomes, emphasizing the role of economic security in maintaining a good quality of life. Perceived health played a crucial role in overall well-being, as individuals who rated their physical health as excellent experienced significantly better quality of life. This suggests that self-assessment of health status is a strong predictor of both physical and mental well-being. Additionally, regional differences in HRQoL were observed, with those residing in the Northeast and Midwest regions reporting better health outcomes. This could be attributed to variations in healthcare access, socioeconomic conditions, environmental factors, or community support systems available in these areas.

Physical activity also emerged as a key contributor to improved HRQoL. Prior research investigated the effects of physical activity on OA patients' physical function and HRQoL using a statewide population-based cohort reported that regular physical activity was related to enhanced physical function and HRQoL among individuals with hip and/or Knee OA [13,42]. The results of this study further highlighted the necessity of managing OA through a multidisciplinary approach that includes lifestyle, psychological, and pharmacological interventions customized to each patient's needs to lessen the negative effects of coexisting mental conditions and, therefore, improve the overall quality of life.

## 4.1. Study strengths and limitations

This study used a large sample size and a nationally representative sample of US adults with OA, which allowed us to obtain a clinically insightful estimate of the relationship between depression and anxiety and HRQoL in adults with OA. However, this study has some limitations. First, the cross-sectional design of our study limits its ability to determine the causal relationship between depression, anxiety, and HRQoL. Second, specific critical confounders were not available on MEPS data and have not been adjusted for in the analysis, including osteoarthritis-related variables (i.e., treatment, the

severity of their illness, disease subtypes, OA-related pain, pain-severity, and pain location (hip vs. knee joint)) which can interfere with a person's ability to function normally and have a negative relationship with HRQoL [17,38,43,44]. These essential confounders were overlooked, which all have a significant role in influencing HRQoL in adults with OA and are crucial to comprehending the intricate relationship between mental health and OA patients' HRQoL. Third, missing these critical OA-related variables in the analysis may have introduced a potential omitted variable bias and weakened the study's findings and conclusion; thus, the results of this study may have overestimated the direct impact of anxiety and depression on HRQoL in persons with osteoarthritis, therefore, adding these crucial factors should be the top priority for future research. However, in this study's analysis compared to previously published research, an adjustment for several essential confounders that can potentially affect the HRQoL, such as social determinants, chronic illnesses, physical activity, health insurance coverage, and perceived health, were included to produce greater confidence in detecting the actual effect of mental health on HRQoL in adults with OA [27–33]. Finally, the fact that data on HRQoL and OA symptoms are mostly self-reported introduces a potential source for biases such as underreporting or misreporting bias.

## 4.2. Clinical practice, policy and public health, and research implications

The clinical practice implication of the results of our research should reassure us that physicians, rheumatologists, and mental health professionals should incorporate routine screening for anxiety and depression in adults with OA, especially when patients or their families report symptoms [45]. This can help minimize suffering, improve symptom management, and enhance HRQoL. Clinicians should adhere to the NICE guidelines on identifying and treating depression, utilizing a combination of pharmacotherapy, cognitive-behavioral therapy (CBT), physical therapies, and structured activities to optimize treatment outcomes [46]. Some research suggested that the use of CBT, education programs, pharmacotherapy, and physical exercise are linked to the short-term reduction of depressed symptoms in older people with OA [35,47]. Besides, healthcare providers should adopt an integrated pain management approach, combining pharmacologic treatments (e.g., simple analgesics, NSAIDs, COX-2 inhibitors) with non-pharmacologic interventions (e.g., movement meditation, acupuncture, and psychological therapy) to reduce pain-related anxiety and depression in OA patients [48–50].

From policy standpoint, our results highlight the importance of integration of mental health screening into routine OA care at primary and specialty healthcare settings [34]. Governments and health insurance providers must enhance access to affordable mental health services by reducing financial barriers for uninsured and underinsured populations. This could include expanding publicly funded mental health programs and increasing coverage for psychological services in health insurance plans. This is especially important due to the high cost of mental health care, as people with mental illnesses have historically faced major barriers to seeking mental health care for the uninsured and even those who are insured [51]. These challenges are especially pronounced among low-income and unemployed populations, who, as demonstrated in our analysis, are disproportionately burdened by comorbid mental health conditions and reduced HRQoL.

The public health implications encourage lifestyle changes like physical activity. Physical activity equivalent to 150 minutes per week of moderate-intensity aerobic activity or 75 minutes per week of vigorous aerobic activity, or a combination of both can have a detrimental positive impact on HRQoL [42,52,53], decreased OA pain [53], improve physical function of OA patients [52,53]. Public health agencies should leverage exercise-based interventions as a cost-effective alternative to antidepressant treatment, given its comparable benefits in reducing depression and anxiety in OA patients [54,55].

Future research should aim to address several limitations identified in this study and further explore the complex interplay between osteoarthritis (OA), mental health, and health-related quality of life (HRQoL). Longitudinal cohort studies are needed to establish causal relationships between anxiety, depression, and changes in HRQoL over time. Additionally, future investigations should incorporate OA specific clinical variables such as pain severity, functional limitation, disease subtype, treatment regimens, and joint location which were not available in the current dataset but are known to substantially influence HRQoL outcomes. Moreover, research focusing on intervention strategies, including integrated care models, physical activity programs, and mental health support tailored to OA patients, is essential to evaluate their

effectiveness in real-world settings. Finally, incorporating patient-reported outcome measures and qualitative insights could enrich understanding of patient priorities and lived experiences, informing the design of more patient-centered care approaches.

## 5. Conclusions

The relationships between anxiety, depression, and a low HRQoL in patients with OA was found in this study. This study also revealed the negative effect of low education and low income on HRQoL while highlighting the advantages of employment and regular exercise. It is crucial for future research to thoroughly evaluate patient-centered clinical approaches care and management policies. This understanding of risk factors that hinder patients with OA, which increase anxiety and depression risk and therefore compromise the HRQoL in individuals with OA, is worthy.

## Author contributions

**Conceptualization:** Monira Alwhaibi, Tariq M. Alhawassi.

**Data curation:** Monira Alwhaibi, Tariq M. Alhawassi.

**Formal analysis:** Monira Alwhaibi.

**Investigation:** Monira Alwhaibi, Tariq M. Alhawassi.

**Methodology:** Monira Alwhaibi, Tariq M. Alhawassi.

**Writing – original draft:** Monira Alwhaibi.

**Writing – review & editing:** Monira Alwhaibi, Tariq M. Alhawassi.

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
