## [Decision Letter · Decision Letter 0]

Apr 16 2025

PONE-D-24-50708Mental Health and Health-Related Quality of Life among Adults with Osteoarthritis: A National Population-Based StudyPLOS ONE

Dear Dr. Alwhaibi,

Thank you for submitting your manuscript to PLOS ONE. After careful consideration, we feel that it has merit but does not fully meet PLOS ONE’s publication criteria as it currently stands. Therefore, we invite you to submit a revised version of the manuscript that addresses the points raised during the review process.

**ACADEMIC EDITOR: Dear Author, please attend to all of the comments provided by the reviewer/s** . ==============================

We look forward to receiving your revised manuscript.

Kind regards,

Zulkarnain Jaafar

Academic Editor

PLOS ONE

Journal Requirements:

3. Please update your submission to use the PLOS LaTeX template. The template and more information on our requirements for LaTeX submissions can be found at http://journals.plos.org/plosone/s/latex .

Reviewers' comments:

Reviewer's Responses to Questions

**Comments to the Author**

1. Is the manuscript technically sound, and do the data support the conclusions?

Reviewer #1: Yes

2. Has the statistical analysis been performed appropriately and rigorously? 

Reviewer #1: Yes

3. Have the authors made all data underlying the findings in their manuscript fully available?

Reviewer #1: Yes

4. Is the manuscript presented in an intelligible fashion and written in standard English?

Reviewer #1: Yes

5. Review Comments to the Author

Reviewer #1: SYNOPSIS:

Mental Health and Health-Related Quality of 1 Life among Adults with Osteoarthritis: A National Population-Based Study

The study is a good addition to knowledge; especially as it utilized a robust data which makes generalizability easier. However, the manuscript will need some fine tuning to improve its quality. If some of the of the recommendations are taken, it might improve the quality of the work.

Details of my observations are attached as comments on the manuscripts. Below are my general observations:

Data analysis:

what determine the inclusion of candidate variables into the multivariable linear regression model?... or all variables were loaded into the omnibus model? ......kindly state this?

Results:

I hope it is the Likelihood Ratio p-value that was reported for the p-values; and indicated as such; as this table is filled with K by K or 2 by K Chi-squares.

All significant results should be reported and not just one from a group of significant variables.

Be specific about the domain you are reporting.

Ensure that all reported regression result ends with a comparison with the reference group.

Discussion:

Discussing an insignificant result while leaving out the significant result might not be okay. E.g gender.

Recommendations:

recommendations should be addressed to specific entities and for specific entities.

References:

Some need re-editing; some revising and incomplete references completed.

6. PLOS authors have the option to publish the peer review history of their article (what does this mean? ). If published, this will include your full peer review and any attached files.

**Do you want your identity to be public for this peer review?** For information about this choice, including consent withdrawal, please see our Privacy Policy .

Reviewer #1: **Yes: ** PHILIP ADEWALE ADEOYE (Jos University Teaching Hospital, Nigeria)

---

## [Author Response · Author response to Decision Letter 1]

7 Mar 2025

Response to Reviewer Comments

We appreciate the reviewer’s thoughtful comments and suggestions, which have helped improve the clarity and quality of our manuscript. Below is our detailed response to each comment:

Data Analysis:

Comment 1: Inclusion of Candidate Variables in Multivariable Linear Regression Model

Response: Thank you for this comment; we have now clarified the selection process of variables in the Methods section. Variables were included based on their theoretical relevance, clinical importance, and prior literature supporting their association with health-related quality of life (HRQoL) in osteoarthritis (OA) patients. Additionally, a bivariate analysis was performed, and only variables with a p-value <0.05 were considered for the multivariable regression model.

Results:

Comment 2: Likelihood Ratio p-value Indication

Response: We have ensured that the reported p-values are based on the likelihood ratio test where appropriate and have explicitly indicated this in the text and tables.

Comment 3: Reporting All Significant Results

Response: We have revised the results section to ensure that all significant findings are reported.

Comment 4: Specificity about Domains Being Reported

Response: Thanks for pointing out, now we have specify whether the reported HRQoL results pertain to the Physical Component Summary (PCS) or the Mental Component Summary (MCS) to enhance clarity.

Comment 5: Comparison with the Reference Group

Response: Thanks for this comment; we have ensured that all the reported regression results include a clear comparison with the reference group.

Discussion:

Comment 6: Addressing Significant vs. Insignificant Results

Response: The discussion has been revised to emphasize significant results while removing discussion on insignificant findings (i.e., gender was previously discussed despite being statistically insignificant; this has now been removed).

Comment 7: Comprehensive Reporting of Key Variables

Response: As per the reviewer recommendations, we have expanded the discussion to cover all significant factors, including perceived health, employment, region, and income, as they were significantly associated with HRQoL. We have also clarified the role of education by providing possible explanations for the observed findings and comparing them with previous studies.

Recommendations:

Comment 8: Specificity in Recommendations

Response: Thanks for the suggestion. Recommendations have been revised to clearly address specific entities such as healthcare providers, policymakers, and public health organizations.

Comment 9: Re-editing and Completing References

Response: The reference section has been thoroughly revised to ensure completeness, correct formatting, and accuracy.

---

## [Decision Letter · Decision Letter 1]

May 23 2025

PONE-D-24-50708R1Mental Health and Health-Related Quality of Life among Adults with Osteoarthritis: A National Population-Based StudyPLOS ONE

Dear Dr. Alwhaibi,

Thank you for submitting your manuscript to PLOS ONE. After careful consideration, we feel that it has merit but does not fully meet PLOS ONE’s publication criteria as it currently stands. Therefore, we invite you to submit a revised version of the manuscript that addresses the points raised during the review process.

**ACADEMIC EDITOR: Dear Author, please attend to all comments provided by the reviewers.**==============================

We look forward to receiving your revised manuscript.

Kind regards,

Zulkarnain Jaafar

Academic Editor

PLOS ONE

Journal Requirements:

Reviewers' comments:

Reviewer's Responses to Questions

**Comments to the Author**

1. If the authors have adequately addressed your comments raised in a previous round of review and you feel that this manuscript is now acceptable for publication, you may indicate that here to bypass the “Comments to the Author” section, enter your conflict of interest statement in the “Confidential to Editor” section, and submit your "Accept" recommendation.

Reviewer #1: All comments have been addressed

Reviewer #2: All comments have been addressed

2. Is the manuscript technically sound, and do the data support the conclusions?

Reviewer #1: Yes

Reviewer #2: Yes

3. Has the statistical analysis been performed appropriately and rigorously? 

Reviewer #1: Yes

Reviewer #2: I Don't Know

4. Have the authors made all data underlying the findings in their manuscript fully available?

Reviewer #1: Yes

Reviewer #2: Yes

5. Is the manuscript presented in an intelligible fashion and written in standard English?

Reviewer #1: Yes

Reviewer #2: Yes

6. Review Comments to the Author

Reviewer #1: The authors have attended to ALL the pervious comments/observations. However, there remain some observations made on the uploaded manuscript as comments. In summary

1. The need to rephrase the title of a table to make it clearer

2. some incorrect narration

3. likely incorrectly stated public health implications.

Once, these concerns are addressed, the work can be accepted for publication.

Thank you.

Reviewer #2: The present study is highly relevant and timely, grounded in a substantial body of literature and organized in a coherent manner. However, there are aspects that can be enhanced to make the work more robust and well-founded. The main areas for potential improvement will be discussed below.

1. Introduction and Theoretical Framework

Although the introduction is well-structured, it could be strengthened by including a more comprehensive review of the existing literature. Presenting previous studies related to the topic, along with their main findings and identified gaps, would contribute to a more solid theoretical foundation.

Additionally, it is essential that fundamental concepts be discussed in greater detail, considering different theoretical and methodological perspectives. A critical analysis of the literature would allow for the identification of alternative approaches and their justification based on clear criteria, such as methodological adequacy, validity of results, and relevance to the investigated context.

2. Justification of the Study

Although the study presents clear objectives, the justification could be enhanced by providing a more detailed explanation of the reasons underlying the choice of the variables investigated. It is necessary to discuss how these variables relate to the research problem and how they contribute to advancing knowledge in the field.

The selection of measurement scales should be justified based on their validity, reliability, and suitability to the study context. Moreover, it is important to address potential biases that may arise from the use of these scales and how they were controlled or minimized in the research.

3. Methodology and Statistical Approach

From a methodological standpoint, the study presents an appropriate design; however, the description of the procedures used could be expanded to provide greater clarity and transparency.

4. Results

The results are clearly presented, with appropriate use of tables, diagrams, and graphs that facilitate the understanding of the findings. However, it is recommended to deepen the discussion of the results by considering different possible interpretations and relating them to previous studies.

In summary, the research has the potential to contribute significantly to the field of study. However, it is essential that the conclusions be based on a rigorous and critical analysis of the findings. It is recommended to discuss not only the direct impacts of the study but also its implications for public policy, clinical practice, and future research. Comparing the findings with other realities, particularly international ones, can provide valuable insights into the relevance of the results.

7. PLOS authors have the option to publish the peer review history of their article (what does this mean? ). If published, this will include your full peer review and any attached files.

**Do you want your identity to be public for this peer review?** For information about this choice, including consent withdrawal, please see our Privacy Policy .

Reviewer #1: **Yes: ** ADEOYE, Philip Adewale (Jos-Nigeria)

Reviewer #2: No

---

## [Author Response · Author response to Decision Letter 2]

30 Apr 2025

We thank the reviewers for their valuable feedback, which has greatly contributed to improving our manuscript. Below, we provide a detailed response to each comment and describe the corresponding revisions made.

Reviewer 1

Comment 1: The need to rephrase the title of a table to make it clearer

Response: Thank you for this suggestion; we have revised the title of the relevant table to improve clarity. Specifically, the title was changed to: Table 1. Socioeconomic and health-related characteristic of study participants across osteoarthritis groups.

Comment 2: some incorrect narration

Response: We carefully reviewed the results and corrected instances of inaccurate or unclear narration (Page 11, Lines 228-230).

Comment 3: likely incorrectly stated public health implications.

Response: We have revised the physical activity recommendations align with current American Heart Association Recommendations for Physical Activity in Adults (Page 19, Lines 371-373).

Reviewer 2

Comment 1: Introduction and Theoretical Framework

Although the introduction is well-structured, it could be strengthened by including a more comprehensive review of the existing literature. Presenting previous studies related to the topic, along with their main findings and identified gaps, would contribute to a more solid theoretical foundation.

Additionally, it is essential that fundamental concepts be discussed in greater detail, considering different theoretical and methodological perspectives. A critical analysis of the literature would allow for the identification of alternative approaches and their justification based on clear criteria, such as methodological adequacy, validity of results, and relevance to the investigated context.

Response: Thank you for your constructive feedback. In response, we have expanded the introduction to include a more comprehensive and critical review of the existing literature. Specifically, we now discuss key methodological limitations in previous studies, such as reliance on self-reported data (Chang et al.), selection bias from online surveys (Tsuji et al.), outdated or single-year data (Kim et al.), small sample sizes (Barbosa et al.), and gender imbalance (Tavares et al.). These additions clarify existing knowledge gaps and support the rationale for our use of a nationally representative U.S. dataset (MEPS) with validated diagnostic codes and HRQoL measures. We believe these enhancements have strengthened the theoretical foundation and contextual relevance of our study (Page 4, Lines 84-103).

Comment 2: Justification of the Study

Although the study presents clear objectives, the justification could be enhanced by providing a more detailed explanation of the reasons underlying the choice of the variables investigated. It is necessary to discuss how these variables relate to the research problem and how they contribute to advancing knowledge in the field.

The selection of measurement scales should be justified based on their validity, reliability, and suitability to the study context. Moreover, it is important to address potential biases that may arise from the use of these scales and how they were controlled or minimized in the research.

Response: We thank the reviewer for this valuable comment. In the revised manuscript, we have expanded the justification for variable selection and clarified how they relate to the research objectives (Page 6-7, Lines 143-153).

Comment 3: Methodology and Statistical Approach

From a methodological standpoint, the study presents an appropriate design; however, the description of the procedures used could be expanded to provide greater clarity and transparency.

Response: We appreciate the reviewer’s valuable feedback. In response, we have revised the Methods section to expand and clarify the study procedures (Page 5, Lines 109-121).

Comment 4: Results

The results are clearly presented, with appropriate use of tables, diagrams, and graphs that facilitate the understanding of the findings. However, it is recommended to deepen the discussion of the results by considering different possible interpretations and relating them to previous studies.

In summary, the research has the potential to contribute significantly to the field of study. However, it is essential that the conclusions be based on a rigorous and critical analysis of the findings. It is recommended to discuss not only the direct impacts of the study but also its implications for public policy, clinical practice, and future research. Comparing the findings with other realities, particularly international ones, can provide valuable insights into the relevance of the results.

Response: Thank you for the constructive feedback. We have expanded the Discussion section to include interpretations, comparisons with international studies (Pages 13-14, Lines 240-253), and broader implications for policy, clinical practice, public and future research (Pges 18-20, Lines 346-389). These enhancements ensure a more rigorous and contextually grounded interpretation of the findings.

---

## [Decision Letter · Decision Letter 2]

Mental Health and Health-Related Quality of Life among Adults with Osteoarthritis: A National Population-Based Study

PONE-D-24-50708R2

Dear Dr. Alwhaibi,

We’re pleased to inform you that your manuscript has been judged scientifically suitable for publication and will be formally accepted for publication once it meets all outstanding technical requirements.

Kind regards,

Zulkarnain Jaafar

Academic Editor

PLOS ONE

Additional Editor Comments (optional):

Reviewers' comments:

Reviewer's Responses to Questions

**Comments to the Author**

1. If the authors have adequately addressed your comments raised in a previous round of review and you feel that this manuscript is now acceptable for publication, you may indicate that here to bypass the “Comments to the Author” section, enter your conflict of interest statement in the “Confidential to Editor” section, and submit your "Accept" recommendation.

Reviewer #1: All comments have been addressed

2. Is the manuscript technically sound, and do the data support the conclusions?

Reviewer #1: Yes

3. Has the statistical analysis been performed appropriately and rigorously? 

Reviewer #1: Yes

4. Have the authors made all data underlying the findings in their manuscript fully available?

Reviewer #1: Yes

5. Is the manuscript presented in an intelligible fashion and written in standard English?

Reviewer #1: Yes

6. Review Comments to the Author

Reviewer #1: The author had adequately attended to my observations and comments in the two rounds of earlier reviews. I recommend this article for acceptance for publication. Thank you for the opportunity to review this work.

7. PLOS authors have the option to publish the peer review history of their article (what does this mean? ). If published, this will include your full peer review and any attached files.

**Do you want your identity to be public for this peer review?** For information about this choice, including consent withdrawal, please see our Privacy Policy .

Reviewer #1: **Yes: ** ADEOYE, Philip Adewale (Jos University Teaching Hospital, Nigeria)

---

## [Editor Report · Acceptance letter]

PONE-D-24-50708R2

PLOS ONE

Dear Dr. Alwhaibi,

I'm pleased to inform you that your manuscript has been deemed suitable for publication in PLOS ONE. Congratulations! Your manuscript is now being handed over to our production team.

Kind regards,

on behalf of

Dr. Zulkarnain Jaafar

Academic Editor

PLOS ONE